# Time-Dependent Reduction of Calcium Oscillations in Adipose-Derived Stem Cells Differentiating towards Adipogenic and Osteogenic Lineage

**DOI:** 10.3390/biom11101400

**Published:** 2021-09-23

**Authors:** Enrico C. Torre, Mesude Bicer, Graeme S. Cottrell, Darius Widera, Francesco Tamagnini

**Affiliations:** 1Stem Cell Biology and Regenerative Medicine Group, School of Pharmacy, University of Reading, Whiteknights, Reading RG6 6LA, UK; e.c.torre2@newcastle.ac.uk (E.C.T.); M.Bicer@pgr.reading.ac.uk (M.B.); 2Neuronal and Cellular Physiology Group, School of Pharmacy, University of Reading, Whiteknights, Reading RG6 6LA, UK; 3Biomedicine West Wing, International Centre for Life, Times Square, Newcastle University, Newcastle upon Tyne NE1 3BZ, UK; 4Department of Bioengineering, Sumer Campus, Abdullah Gül University, Kayseri 38080, Turkey; 5Cellular and Molecular Neuroscience, School of Pharmacy, University of Reading, Reading RG6 6LA, UK; g.s.cottrell@reading.ac.uk

**Keywords:** calcium oscillations, osteogenic differentiation, adipogenic differentiation, mesenchymal stem cells, adipose-derived stem cells

## Abstract

Adipose-derived mesenchymal stromal cells (ASCs) are multipotent stem cells which can differentiate into various cell types, including osteocytes and adipocytes. Due to their ease of harvesting, multipotency, and low tumorigenicity, they are a prime candidate for the development of novel interventional approaches in regenerative medicine. ASCs exhibit slow, spontaneous Ca^2+^ oscillations and the manipulation of Ca^2+^ signalling via electrical stimulation was proposed as a potential route for promoting their differentiation in vivo. However, the effects of differentiation-inducing treatments on spontaneous Ca^2+^ oscillations in ASCs are not yet fully characterised. In this study, we used 2-photon live Ca^2+^ imaging to assess the fraction of cells showing spontaneous oscillations and the frequency of the oscillation (measured as interpeak interval—IPI) in ASCs undergoing osteogenic or adipogenic differentiation, using undifferentiated ASCs as controls. The measurements were carried out at 7, 14, and 21 days in vitro (DIV) to assess the effect of time in culture on Ca^2+^ dynamics. We observed that both time and differentiation treatment are important factors associated with a reduced fraction of cells showing Ca^2+^ oscillations, paralleled by increased IPI times, in comparison with untreated ASCs. Both adipogenic and osteogenic differentiation resulted in a reduction in Ca^2+^ dynamics, such as the fraction of cells showing intracellular Ca^2+^ oscillations and their frequency. Adipogenic differentiation was associated with a more pronounced reduction of Ca^2+^ dynamics compared to cells differentiating towards the osteogenic fate. Changes in Ca^2+^ associated oscillations with a specific treatment had already occurred at 7 DIV. Finally, we observed a reduction in Ca^2+^ dynamics over time in untreated ASCs. These data suggest that adipogenic and osteogenic differentiation cell fates are associated with specific changes in spontaneous Ca^2+^ dynamics over time. While this observation is interesting and provides useful information to understand the functional correlates of stem cell differentiation, further studies are required to clarify the molecular and mechanistic correlates of these changes. This will allow us to better understand the causal relationship between Ca^2+^ dynamics and differentiation, potentially leading to the development of novel, more effective interventions for both bone regeneration and control of adipose growth.

## 1. Introduction

Mesenchymal stem cells (MSCs) are multipotent, fibroblast-like cells that can be readily obtained from various adult tissues including the bone marrow, adipose tissue, and peripheral blood as well as from prenatal tissues such as amniotic fluid, umbilical cord, and placenta. However, as they are derived from the mesoderm, their differentiation spectrum is limited to mesenchymal derivatives such as bone, fat, and cartilage (see [1] for review). Nevertheless, as of July 2021, over 1200 clinical trials involving MSCs have been registered on the ClinicalTrials.gov database with wide indication profiles including diabetes, cardiovascular disorders, in addition to musculoskeletal symptoms. Indeed, transplantation of MSCs has been shown to alleviate symptoms of a broad spectrum of conditions including liver cirrhosis (affected germ layer: endoderm) [2], severe ischaemic heart failure (affected germ layer: mesoderm) [3], and progressive multiple sclerosis (affected germ layer: ectoderm) [4]. Despite the lack of differentiation beyond the mesodermal germ layer boundary, their surprising ability to improve all these conditions can be explained by bystander effects mediated by secretion of paracrine factors [5,6,7,8,9,10].

Adipose-derived stem cells (ASCs) are tissue-resident ASCs that can be isolated from lipo-aspirates [11]. Notably, their differentiation potential towards the mesenchymal fate is comparable to that of ASCs from other organs including the bone marrow [12,13]. Moreover, like bone marrow ASCs, they are also able to contribute to regeneration of various tissues via paracrine effects [13,14,15].

Intracellular Ca^2+^ is a well-known second messenger, involved in downstream signalling cascades and numerous cell functions, including proliferation, differentiation, and apoptosis [16]. It has been demonstrated that oscillations of intracellular Ca^2+^ are associated and causally related to the ability of single cells to process environmental inputs, encode information, and generate an adequate behavioural response: this is known as Ca^2+^ encoding or the Ca^2+^ code [17,18].

ASCs have been reported to exhibit spontaneous Ca^2+^ oscillations and this has been linked to the activity of Ca^2+^ channels and pumps on both the plasma membrane and the endoplasmic reticulum [19,20]. The Ca^2+^ oscillations are triggered by an ATP-dependent autocrine and paracrine signalling loop and subsequently sustained and amplified by inositol triphosphate (IP_3_)-dependent release from internal stores and by an activation of plasma membrane Ca^2+^ channels [21]. The downstroke phase of the Ca^2+^ oscillation is then mediated by the sarco/endoplasmic reticulum Ca^2+^-ATPase pumps (SERCA) and the plasma membrane Ca^2+^-ATPase pumps (PMCA), but not the Na^+^/Ca^2+^ exchanger [22]. Importantly, the association between specific differentiation pathways and the features of the intracellular Ca^2+^ oscillation, such as the fraction of cells showing such activity and the frequency of the event, is still not fully understood. Understanding and establishing the association between the features of the intracellular Ca^2+^ oscillation and the cell fate can be a powerful tool for screening libraries of stem cell-derived cytotypes and will allow us to acquire further insights on the functional correlates of the differentiation process. In fact, measuring Ca^2+^ dynamics in differentiating cells may become an alternative or complementary method to measuring the specific morphological and molecular markers which are currently used to establish the specialisation of a differentiated stem cell-derived cytotype and assess their functional maturity [23]. However, we are still lacking information allowing us to associate specific intracellular Ca^2+^ patterns with different cell fates. One complication which might hinder the deciphering of the Ca^2+^ code, is that intracellular Ca^2+^ dynamics are not only affected by the differentiation protocol, but also by time, especially in undifferentiated cells [24].

In this study, we assessed how time and adipogenic and osteogenic differentiation treatments affect intracellular Ca^2+^ dynamics in ASCs. We measured the effects of time (7 days in vitro, DIV vs. 14 DIV vs. 21 DIV) and the differentiation protocol (adipogenic vs. osteogenic) on the fraction of differentiating ASCs exhibiting at least one spontaneous intracellular Ca^2+^ oscillation and on the interpeak interval (IPI) between oscillations.

This study represents a further step in characterising the association between intracellular Ca^2+^ oscillations and differentiation status of ASCs.

## 2. Materials and Methods

### 2.1. Cultivation of Human ASCs

Human ASCs from three non-diabetic adult donor lipoaspirates characterised at passage 1 were obtained from Lonza (Slough, UK). All ASCs have been characterised immunocytochemically and by tri-lineage differentiation assay as detailed by the manufacturer and as recommended by The International Society for Cellular Therapy [25]. All cells were used between passages 7 and 11. For biological replicates, ASCs within a range of 3 passages were used. ASCs have been characterised immunocytochemically and by tri-lineage differentiation assay as detailed by the manufacturer and as recommended by The International Society for Cellular Therapy [25]. All cells were used on or before passage 10. ASCs were cultivated in a standard medium composed of DMEM high glucose, 100U penicillin with 100 μg/mL streptomycin, 2 mM l-glutamine (all from Sigma-Aldrich, Gillingham, UK), 20% *v/v* heat-inactivated fetal bovine serum (FBS) (Sigma-Aldrich, lot: 8204188981) and 5 ng/mL fibroblast growth factor (FGF2) (Promega, Southampton, UK) at 37 °C and 10% CO_2_. Medium was changed every 2–3 days.

### 2.2. Osteogenic Differentiation

ASCs were seeded at a density of 6 × 10^5^ mL on 13 mm glass cover slips (Agar Scientific, Stanstead, UK) pre-treated with nitric acid. After 72 h, standard medium was replaced by StemPro^®^ Osteocyte basal medium supplemented with StemPro^®^ osteogenesis supplement according to manufacturer’s instructions (Life Technologies, Thermo Fisher Scientific, Loughborough, UK). Cells were cultivated for up to 21 days in a humidified incubator (Binder APT.lineTM C150, Binder, Tuttlingen, Germany) at 37 °C and 5% CO_2_. Media were changed every three days. Osteogenic differentiation was assessed by Alizarin Red S staining at day 21 as described [26]. Briefly, ASCs differentiated for 21 days were fixed for 15 min using 4% PFA followed by three wash steps using PBS with 5 min per wash step. Calcium deposition was visualised by staining the cells with 1% Alizarin Red S in double-deionised water (ddH_2_O, Sigma-Aldrich) at pH 4.3 for 5 min at room temperature followed by imaging using a Nikon A1R inverted confocal microscope (Nikon, Kingston upon Thames, UK).

### 2.3. Adipogenic Differentiation

ASCs were seeded at a density of 6 × 10^5^ mL on 13 mm glass cover slips (Agar Scientific) pre-treated with nitric acid. After 72 h, standard medium was replaced by StemPro^®^ adipocyte differentiation basal medium supplemented with StemPro^®^ adipogenesis supplement (Life Technologies, Thermo Fisher Scientific). After 21 days, differentiation was assessed by Oil RedO staining as described earlier [27].

### 2.4. Calcium Imaging

Calcium imaging experiments were carried out at 7, 14, and 21 DIV since the application of the differentiation protocol. For each timepoint we have measured the effect of three separate differentiation treatments: adipogenic, osteogenic, and control. To perform calcium imaging experiments, the cells were loaded with Fluo-4, a non-ratiometric calcium sensitive fluorescent probe. Cells were washed with PBS (Sigma-Aldrich) and treated with a loading solution containing Hanks’ Balanced Salt Solution (1×), 0.5 mM MgCl_2_, 1.3 mM CaCl_2_, 25 mM HEPES pH 7.4, 2.5 mM Probenecid (Sigma-Aldrich), and 2.5 µM Fluo-4 AM cell permeant (Invitrogen). Following a 45 min incubation at 37 °C, each sample was washed with PBS (2 × 1 min). Subsequently, the samples were stored in an incubator (37 °C, 5% CO_2_) in the loading solution described above (without Fluo-4 AM) until the experiment was carried out (between 15 min and 2 h).

Individual samples were placed into a submerged style recording chamber and continuously perfused (1–3 mL/min) with loading solution (without Fluo-4 AM) and kept at 32 ± 1 °C with a continuous flow temperature controller (Scientifica LTD, Uckfield, UK). Cells were visualised with a 40× water dipping, long-working distance objective, using infrared, differential interference contrast microscopy. A field of view containing at least 10 cells was chosen. The scope was subsequently switched to 2-photon mode. Excitation of the fluorophore was elicited by a Titanium-Sapphire (Ti-Sa), pulsating IR laser (MaiTai, Spectra-Physics, Didcot, UK) with an 810 nm wavelength. The intensity of the laser beam was modulated with a motorised beam attenuator (average laser power 4–5 mW). Emitted fluorescence was detected with two fast oscillating mirrors connected to a Galvo-Galvo system (Scientifica Ltd., Uckfield, UK). The signal to noise ratio increased using photomultiplier tubes (average of 800 arbitrary units) detecting red and green photons. Regions of interest (ROI, 412 µm × 412 µm; 3 for each coverslip) were scanned for 20 min resulting in 912 frames (0.76 frame/s).

### 2.5. Image Processing and Analysis

Analysis of the 2-photon microscopy images was performed using the open source image processor Fiji ImageJ [28]. Over 3750 cells were counted, and over 1440 cells were singularly analysed for measuring the IPI.

### 2.6. Cell Activity

Cells showing at least one oscillation were selected and categorised as “active”. Appendix A shows representative traces of the oscillations observed in each condition. Cells without peaks were classified as “non active” The activity in percentage was calculated as:$$Activity\left(\%\right)=\frac{Active\;cells\;in\;ROI\times 100\;}{Total\;cell\;number\;in\;ROI}$$
The ROIs were considered the source of variability as no significant discrepancy between ROI and single coverslips as sources of variability was observed (data not shown). Data are presented as Mean ± SEM (standard error of the mean).

### 2.7. Interpeak Interval

Cells showing at least two fluorescence peaks were selected and their boundaries were drawn with Fiji’s plug-in “freehand selection”. The peak was the local maximum point of the oscillation, detected with the peak detection tool from Origin 2018 (OriginLab, Stoke Mandeville Bucks, UK). The arbitrary fluorescence data obtained were exported to Origin 2018 (OriginLab, Stoke Mandeville Bucks, UK) for further analysis. Each peak’s timepoint and height was measured with the software Originlab 2018 (Originlab, Stoke Mandeville Bucks, UK). The IPI was measured as the difference between the 2nd and the 1st peak by arithmetic subtraction of the respective time of the peak.

### 2.8. Statistical Analysis

Statistics were performed using Excel and SPSS, as required. We tested the effect of time in vitro (7 DIV, 14 DIV, 21 DIV) and the respective treatment (adipogenic, osteogenic, control) on two separate dependent variables: (1) fraction of active cells, and (2) IPI. A 2-way Analysis of Variance (ANOVA) followed by pairwise comparisons (timepoints within treatment and treatment within timepoint) consisting of post-hoc t-tests with Bonferroni correction (confidence interval 95%) was used. *p* < 0.05 was the threshold adopted to reject the null hypothesis.

## 3. Results

The aim of this study was to measure changes of intracellular Ca^2+^ dynamics in ASCs undergoing osteogenic and adipogenic differentiation, respectively, at 7, 14 and 21 DIV.

First, we confirmed that ASCs treated with osteogenic differentiation medium showed extracellular, insoluble deposits of CaPO_4_ as shown by Alizarin RedS staining at 21 DIV (Figure 1A). Using 2-photon microscopy, we have also identified fluorescent deposits, which likely correspond to deposits of CaPO_4_, early as 7 DIV (Figure 1C, bottom panels: note the extracellular, fluorescent deposits). This observation confirmed that ASCs differentiated into osteocytes as early as 7 DIV. In addition, ASCs differentiated into adipocytes after 14 DIV. ASCs treated with adipogenic differentiation medium showed intracellular lipid droplets, positive for Oil Red O (Figure 1B) at 21 DIV. However, 2-photon microscopy revealed the presence of vacuoles already at 14 DIV but not at 7 DIV (Figure 1C, middle panels: note the large vacuoles). Finally, ASCs cultivated in standard medium did not show positive stain to neither Alizarin RedS nor Oil Red O staining (Figure 1A,B) and 2-photon microscopy did not reveal any deposit nor vacuole at any of the timepoints investigated (Figure 1C, top panels).

Subsequently, we assessed the effect of time and differentiation treatment on the Ca^2+^ dynamics. These were measured as (1) the fraction of cells showing at least one oscillation among the cells counted in each ROI (Figure 2) and (2) the IPI between two subsequent oscillations within those cells showing oscillations (Figure 3).

### 3.1. Effect of Time on Ca^2+^ Aynamics in ASCs Undergoing Separate Differentiation Aaths

Time in vitro had a significant effect on the fraction of cells showing Ca^2+^ oscillations (Figure 2A, Two-way ANOVA, F = 27.99, *p* < 0.001, Source of variability: DIV) and on the length of the IPI in active cells (i.e., those cells showing Ca^2+^ oscillations. Figure 3A, Two-way ANOVA, F = 23.02, *p* < 0.001, Source of variability: DIV). Over 90% of all untreated ASCs showed Ca^2+^ oscillations at both 7 (*n* = 9) and 14 DIV (*n* = 4; *p* > 0.05), with a drop to 40% at 21 DIV (*n* = 4; Figure 2A, left panel, *p* < 0.001). This observation was paralleled by the increase of the IPI at 21 days (*n* = 36; Figure 3A, left panel) compared to both 14 DIV (*n* = 36, *p* = 0.016) and 7 DIV (*n* = 79; *p* = 0.001). However, there was no difference in the IPI between 14 DIV and 7 DIV within control cells (*p* = 1). Less than 20% of cells undergoing adipogenic differentiation showed Ca^2+^ oscillations and time had no effect on the percentage of cells exhibiting activity at 7 DIV (*n* = 8), 14 DIV (*n* = 7) and 21 DIV (*n* = 6; Figure 2A, middle panel, Non-significant—NS). This observation was not paralleled by the IPI (Figure 3A, middle panel): in fact, cells undergoing adipogenic treatment showed an increase of the IPI from 7 DIV (*n* = 10) to 14 DIV (*n* = 4; *p* < 0.0001), followed by a subsequent decrease of the IPI at 21 DIV (*n* = 5; *p* = 0.03) in comparison to the 14 DIV timepoint. In addition, the IPI at 21 DIV was significantly longer than the one at 7 DIV (0.017). Finally, circa 60% of cells undergoing osteogenic differentiation showed Ca^2+^ oscillations at 7 DIV (*n* = 9). This value fell to around 40% at 14 DIV (*n* = 6; *p* < 0.001) to then return to the levels observed at 7 DIV, at 21 DIV (*n* = 4; Figure 2A, right panel). In addition, the IPI within the cells undergoing osteogenic differentiation followed a trend in time similar to the one observed in cells undergoing adipogenic treatment (Figure 3A, right panel). In fact, in these cells the IPI increased between 7 DIV and 14 DIV (*p* = 0.005) and 21 DIV (*p* = 0.011), with no difference between 14 and 21 DIV (*p* = 1).

For a detailed list of the pairwise comparisons between timepoints within each treatment, see Table 1.

### 3.2. Effect of Osteogenic and Adipogenic Aifferentiation on Ca^2+^ Aynamics in ASCs at Aifferent Timepoints

The type of differentiation treatment had a significant effect on the fraction of cells showing Ca^2+^ oscillations (Figure 2B, Two-way ANOVA, F = 369.61, *p* < 0.001, Source of variability: differentiation treatment) and on the IPI between Ca^2+^ oscillations in active cells (Figure 3B, Two-way ANOVA, F = 21.75, *p* < 0.001, Source of variability: differentiation treatment). Post-hoc *t*-tests with Bonferroni correction allowed us to conclude that adipogenic differentiation resulted in a reduced portion of active cells, compared to both controls (*p* < 0.001) and cells undergoing osteogenic differentiation (*p* < 0.001) at all the considered timepoints (Figure 2B). Cells undergoing osteogenic differentiation exhibited reduced activity compared with controls, both at 7 DIV and 14 DIV (Figure 2B, left and middle panel, *p* < 0.001 at both timepoints), but higher activity at 21 DIV (Figure 2B, right panel, *p* < 0.001). These observations were partially paralleled by differentiation-dependent changes of the IPI at each timepoint. First, the treatment did not affect the IPI at 7 DIV. Secondly, cells undergoing adipogenic differentiation exhibited longer IPIs compared with controls and osteogenic cells at 14 DIV (*p* < 0.001). However, at 21 DIV, cells undergoing adipogenic treatment showed longer IPIs compared to those undergoing osteogenic treatment (*p* = 0.006) but not against controls (*p* = 0.4).

For a detailed list of the pairwise comparisons between timepoints within each treatment, see Table 2.

Finally, we observed that time and differentiation treatments interact to affect the fraction of cells showing Ca^2+^ oscillations (Two-way ANOVA, F = 46.62, *p* < 0.001, Source of variability: Time × Differentiation treatment). Finally, time and differentiation treatment interact in affecting the length of the IPI of Ca^2+^ oscillations (Two-way ANOVA, F = 7.28, *p* < 0.001, Source of variability: Time × Differentiation treatment).

## 4. Discussion

We investigated the effect of adipogenic and osteogenic differentiation treatments on the intracellular Ca^2+^ dynamics of ASCs at three different timepoints in vitro.

We noted that the different differentiation protocols led to the development of structural markers of differentiation. For example, we observed extracellular Ca^2+^ deposits positive to Alizarin Red S for the osteogenic treatment. 2-photon imaging revealed, in a nonspecific manner, the presence of Ca^2+^ deposits at 7 DIV in ASCs treated with the osteogenic medium. This increase in Ca^2+^ deposition was paralleled by an increased expression of mRNA for osteocalcin and osteopontin, molecular markers for osteogenic differentiation [26]. However, this observation was in contrast with a previous research work, which showed no increase in mRNA for osteocalcin or osteopontin in ASCs treated with osteogenic medium (21 DIV) [29]. Such discrepancy may be due methodological differences between the experimental conditions, such as the composition of the differentiation media. This, in turn, may underlie the presence of reservoirs of bone marker mRNAs in the cytoplasm of ASCs, which are translated in the presence of specific chemical and/or physical cues. Therefore, further investigation is needed to clarify the effect of osteogenic differentiation on molecular markers and its causal relationship with calcium deposition and oscillation. We have observed intracellular vacuoles positive to OilRedO, indicating adipogenic differentiation, for the sample treated with adipogenic differentiation medium, at 21DIVs. Once again, 2-photon imaging revealed presence of vacuoles already at 14 DIV in the sample treated with adipogenic medium.

Almost all the undifferentiated ASCs exhibited spontaneous Ca^2+^ oscillations (Figure 2A, left panel) with relatively short IPIs (Figure 3A, left panel), at 7 and 14 DIV. However, at 21 DIV less than 50% of ASCs exhibited oscillations (Figure 2A, left panel) and the length of the IPI increased by almost 100% compared to the 7 DIV timepoint (Figure 3A, left panel). This suggests that time has a significant effect in reducing the spontaneous Ca^2+^ dynamics in untreated ASCs. While it has already been shown that ASCs possess spontaneous Ca^2+^ oscillations (19-22), a time-dependent reduction in Ca^2+^ dynamics has not been previously shown. Therefore, future work aimed at establishing a correlation (and possibly a causal relation) between the properties of Ca^2+^ oscillations and cell differentiation, will be important to account for a possible biasing action of time on spontaneously occurring Ca^2+^ oscillations.

We observed that the adipogenic differentiation treatment generally reduced the Ca^2+^ dynamics, as less than 20% of cells exhibited Ca^2+^ oscillations at any measured timepoint (Figure 2A, middle panel) and those with a Ca^2+^ oscillations had IPIs exceeding 200 s (Figure 3A, middle panel). This observation is novel, as previous studies on the role of intracellular Ca^2+^ oscillations in differentiating adipocytes, mostly focused on evoked rather than spontaneous Ca^2+^ transients [30]. While multiple studies have shown the pivotal role of Ca^2+^ signalling in adipogenesis [30,31,32], our data show a markedly significant decrease in the fraction of oscillating cells as early as 7 DIV from the initiation of the differentiation protocol (Figure 2B). A possible explanation of this discrepancy is that while the increase in intracellular Ca^2+^ might be the trigger to adipogenic differentiation, we measured a decrease in Ca^2+^ dynamics following the incubation of cells with an adipogenic differentiation medium. Although ASCs (at 7 DIV) undergoing adipogenic differentiation lacked vacuoles, indicating an undifferentiated state (Figure 1C, middle panels), there was a decreased fraction of cells with Ca^2+^ activity (Figure 2B). In addition, IPIs increased at 14 DIV only (Figure 3B), in comparison to time-matched controls and ASCs undergoing osteogenic differentiation. This observation is interesting because it suggests that the differentiation-dependent reduction in Ca^2+^ activity may be an early, functional marker of adipogenesis, preceding other structural and molecular changes typical of adipocytes, such as lipid-filled vacuoles. Analogously, the fraction of active ASCs undergoing adipogenic differentiation was significantly reduced at 14 and 21 DIV, in comparison to untreated ASCs and those undergoing osteogenic differentiation. However, the ASCs undergoing adipogenic differentiation had longer IPIs, in comparison to the other treatments only at 14 DIV and 21 DIV. We postulate that this is likely due to the effect of time in vitro reducing spontaneous Ca^2+^ dynamics in untreated ASCs.

We also examined the effect of osteogenic differentiation on Ca^2+^ dynamics. Overall, osteogenic differentiation is associated with a decrease of the fraction of active cells, (Figure 2B) and an increase of the length of the IPI (Figure 3B) compared to untreated ASCs. The reduction in Ca^2+^ dynamics associated with osteogenic differentiation over time that we observed is consistent with a previous study [33]. Interestingly, in our experimental set up we observed a reduction in the fraction of active cells at 14 DIV, followed by a recovery at 21 DIV (Figure 2A). This biphasic change deserves further investigation. The fraction of active cells undergoing osteogenic differentiation was consistently higher than that of ASCs undergoing adipogenic differentiation (Figure 2B) but lower than controls at 7 and 14 DIV. However, at 21 DIV, cells undergoing osteogenic differentiation show less activity in comparison to controls. The IPIs, while being consistently shorter than the ones of adipogenic cells (Figure 3B), were not significantly different to control values, except at 21 days, where the osteogenic values were slightly but significantly lower than the controls (Figure 3B). This latter observation may be due to the effect of time reducing Ca^2+^ dynamics in untreated ASCs, but not in ASCs undergoing osteogenic differentiation. This is interesting because the manipulation of Ca^2+^ signalling has been proposed as a pivotal mechanism for the induction of osteogenic differentiation [20,34,35], leading to the development of electrical stimulation-based systems to promote osteogenic differentiation, as a possible tool for bone regeneration [26,27]. As we have shown that spontaneous Ca^2+^ oscillations in untreated ASCs are reduced over time, it is plausible to speculate that external stimuli promoting sustained Ca^2+^ oscillations may be responsible for osteogenic differentiation. However, it cannot be excluded that the maintenance of Ca^2+^ oscillations over time may be an epiphenomenon following and not causing osteogenic differentiation.

Changes in Ca^2+^ oscillation properties have been previously associated with separate differentiation fates [35,36]. In addition, it has been shown by Sun et al. that human MSCs derived from bone marrow can undergo osteogenic differentiation following the application of both electrical and chemical differentiation-inducing stimuli. Their study illustrated that bone marrow-derived MSCs undergoing osteogenic differentiation showed a progressive reduction in the number of intracellular [Ca^2+^] spikes as differentiation progressed [35]. While our study partially confirms a time-dependent reduction of Ca^2+^ dynamics over time, this occurs in in adipose-derived, rather than bone marrow-derived MSCs undergoing osteogenic differentiation. In addition, we have also observed a similar reduction in active control cells. Furthermore, Sun et al. suggested that the changes in the intracellular Ca^2+^ dynamics of MSCs differentiating into osteocytes may be both a consequence and a cause of the differentiation [35]. More specifically, it has also been shown that Ca^2+^ oscillations in human MSCs are generated by a combination of extracellular and intracellularly stored Ca^2+^. Extracellular Ca^2+^ enters the cell via the activation of a plasma membrane voltage-gated Ca^2+^ channels, while intracellularly stored one can be released via the activation of IP_3_ sensitive Ca^2+^ channels on the endoplasmic reticulum, which can be activated as a downstream signalling event following the binding of a ligand to integrins and G_q_-coupled receptors [35,37]. Therefore, it is possible to infer that specific pattern of electrical stimulation may lead to specific differentiation fates (such as adipogenic and osteogenic) by modulating the properties of the intracellular Ca^2+^ dynamics. It is still to be established; however, how and if the modulation of Ca^2+^ waves is causative of the differentiation process or an epiphenomenon. Future work aimed at establishing a correlation (and possibly a causal relation) between the properties of Ca^2+^ oscillations and cell differentiation, will be important to account for a possible biasing action of time on spontaneously occurring Ca^2+^ oscillations.

## 5. Conclusions

Our main observations are that spontaneously occurring Ca^2+^ oscillations in ASCs are reduced over time and by adipogenic differentiation, while they are maintained over time by osteogenic differentiation. This observation is important because some changes in Ca^2+^ dynamics occur in both the adipogenic and osteogenic treatments at 7 DIV, before the structural markers of adipocytes (lipid-filled vacuoles) appear. For this reason, Ca^2+^ dynamics should be further investigated as a potential early marker of differentiation. However, further research work is needed to assess the causal relationship between Ca^2+^ oscillation dynamics and cell fate.

## Figures and Tables

**Figure 1 biomolecules-11-01400-f001:**
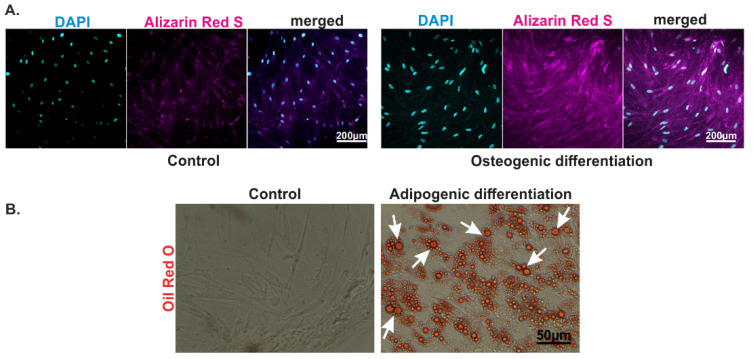
Osteogenic and adipogenic differentiation of ASCs. ASCs were cultivated in standard control medium, or the respective differentiation medium for 7–21 days, followed by staining with DAPI, Alizarin RedS or OilRedO. (**A**) Alizarin RedS staining (magenta) revealed CaPO_4_ deposition in cells exposed to the osteogenic differentiation medium, whereas no Alizarin RedS signal was detected in control cells, at 21 DIV. Scale bar: 200 µm. (**B**) ASCs were exposed to adipogenic differentiation medium or standard medium (control) for 21 days followed by staining of lipid droplets using Oil Red O. Note: Oil Red O-stained lipid droplets in the cells treated with adipogenic differentiation medium (arrows) which was absent under control conditions, at 21 DIV. Scale bar: 50 µm (**C**) Differentiated and undifferentiated ACSs were loaded with Fluo-4 AM followed by live cell imaging using a 2-photon microscope. Ca^2+^ imaging was carried out. The top row shows untreated ASCs. The middle row shows adipogenic differentiation medium-treated ASCs: note the large vacuoles at 14 DIV and 21 DIV. The bottom row shows osteogenic differentiation medium-treated ASCs: note the extracellular deposits at 7 DIV, 14 DIV and 21 DIV. Scale bar: 100 µm.

**Figure 2 biomolecules-11-01400-f002:**
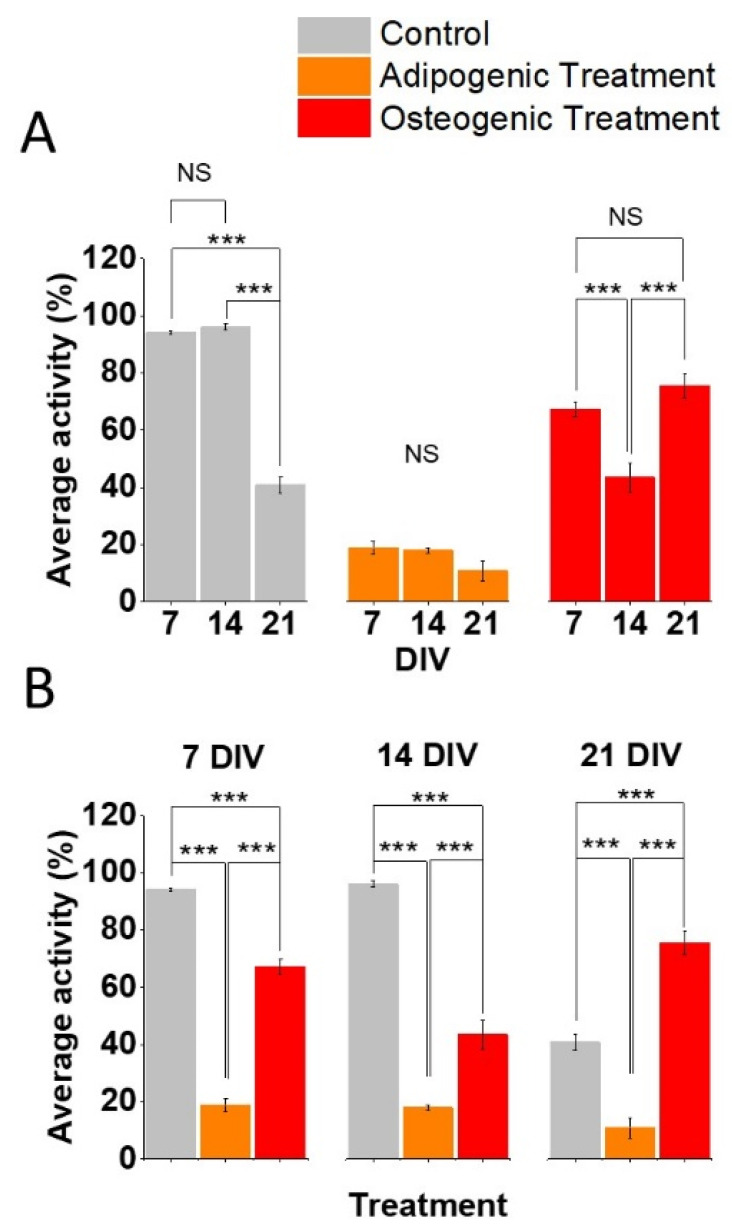
Both time and differentiation treatment affect the fraction of cells with Ca^2+^ oscillations. Untreated ASCs exhibited a significantly lower activity at 21 DIV ((**A**), **left**). Adipogenic treatment is associated with a strong reduction of the activity for all the timepoints ((**A**), **middle**) while the osteogenic treatment shows a decrease at 14 DIV and a subsequent increase at 21 DIV ((**A**), **right**). Cells undergoing adipogenic and osteogenic differentiation showed decreased activity, in comparison to controls, at 7 DIV ((**B**), **left**) and 14 DIV ((**B**), **middle**). However, the ASCs treated with osteogenic differentiation medium showed increased activity at 21 DIV vs. both the cells treated with adipogenic differentiation medium and control ones, while the adipogenic one was still lower than controls ((**B**), **right**). Multiple pairwise comparisons were carried out with *t*-test with Bonferroni correction: *** = *p* ≤ 0.001.

**Figure 3 biomolecules-11-01400-f003:**
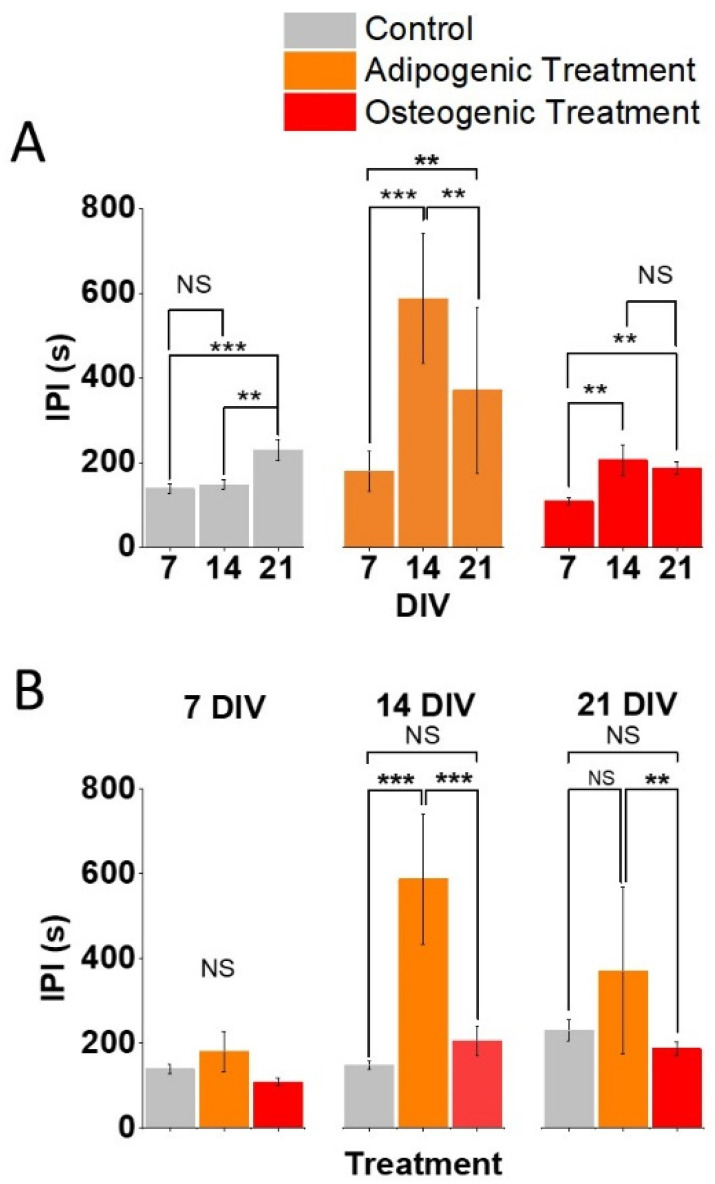
Both time and differentiation treatment affect the IPIs duration of spontaneous Ca^2+^ oscillations. Untreated ASCs show increased interpeak intervals (IPI) at 21 days ((**A**), **left**), whereas cells exposed to adipogenic differentiation medium show an increase at 14 DIV and 21 DIV in comparison to 7 DIV, with a reduction at 21 DIV in comparison to 14 DIV ((**A**), **middle**). The osteogenic samples show an increase in IPI at 14 DIV and 21 DIV in comparison to 7 DIV, but no difference was observed between 14 DIV and 21 DIV ((**A**), **right**). No difference in the IPI was observed across treatments at 7 DIV ((**B**), **left**). The adipogenic sample showed IPIs which were longer than those of controls and the osteogenic sample at 14 DIV ((**B**), **centre**). Finally, at 21 DIV the adipogenic sample showed longer IPIs in comparison to those observed in the osteogenic one, but no difference was observed between controls and either treatment ((**B**), **right**). Multiple pairwise comparisons were carried out with t-test with Bonferroni correction: *** = *p* ≤ 0.001, ** = *p* < 0.05.

**Table 1 biomolecules-11-01400-t001:** Pairwise comparisons between timepoints within each treatment. *p* values have been calculated using unpaired Student’s *t*-tests with Bonferroni correction.

Treatment	DIV (I)	DIV (J)	Fraction of Active Cells		Interpeak Intervals	
			Effect Size (Mean Difference: I–J)	*p* (Bonferroni Corrected)	Effect Size (Mean Difference: I–J)	*p* (Bonferroni Corrected)
control	7	14	−1.985	1.000	−9.020	1.000
		21	53.274	<0.001	−91.319	0.001
	14	7	1.985	1.000	9.020	1.000
		21	55.258	<0.001	−82.298	0.016
	21	7	−53.274	<0.001	91.319	0.001
		14	−55.258	<0.001	82.298	0.016
adipogenic	7	14	0.932	1.000	−407.050	0.000
		21	7.944	0.130	−190.274	0.017
	14	7	−0.932	1.000	407.050	0.000
		21	7.013	0.244	216.776	0.030
	21	7	−7.944	0.130	190.274	0.017
		14	−7.013	0.244	−216.776	0.030
osteogenic	7	14	23.905	<0.001	−96.869	0.005
		21	−8.124	0.187	−77.912	0.011
	14	7	−23.905	<0.001	96.869	0.005
		21	−32.029	<0.001	18.957	1.000
	21	7	8.124	0.187	77.912	0.011
		14	32.029	<0.001	−18.957	1.000

**Table 2 biomolecules-11-01400-t002:** Pairwise comparisons between treatments within each timepoint. *p* values have been calculated using unpaired Student’s *t*-tests with Bonferroni correction.

DIV	Treatment (I)	Treatment (J)	Fraction of Active Cells		Interpeak Intervals	
			Effect Size (Mean Difference: I–J)	*p* (Bonferroni Corrected)	Mean Difference (I–J)	*p* (Bonferroni Corrected)
7	control	adipogenic	75.226	<0.001	−41.435	0.965
		osteogenic	26.774	<0.001	30.031	0.510
	adipogenic	control	−75.226	<0.001	41.435	0.965
		osteogenic	−48.451	<0.001	71.466	0.287
	osteo	control	−26.774	<0.001	−30.031	0.510
		adipogenic	48.451	<0.001	−71.466	0.287
14	control	adipogenic	78.142	<0.001	−439.465	<0.001
		osteogenic	52.664	<0.001	−57.818	0.236
	adipogenic	control	−78.142	<0.001	439.465	<0.001
		osteogenic	−25.478	<0.001	381.647	<0.001
	osteogenic	control	−52.664	<0.001	57.818	0.236
		adipogenic	25.478	<0.001	−381.647	<0.001
21	control	adipogenic	29.896	<0.001	−140.390	0.056
		osteogenic	−34.623	<0.001	43.437	0.418
	adipogenic	control	−29.896	<0.001	140.390	0.056
		osteogenic	−64.520	<0.001	183.827	0.006
	osteogenic	control	34.623	<0.001	−43.437	0.418
		adipogenic	64.520	<0.001	−183.827	0.006

## Data Availability

Data can be provided upon request.

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
