# Peer review of "Time-Dependent Reduction of Calcium Oscillations in Adipose-Derived Stem Cells Differentiating towards Adipogenic and Osteogenic Lineage"

_biomolecules, 2021, doi:10.3390/biom11101400_

Round 1

Reviewer 1 Report

Torre and colleagues tried to clarify time-course Ca oscillations change in differentiation.

Their topic is original, however, there are some critical points in analysis of oscillations.

The authors did not show original oscillation wave in each cell. Readers cannot judge whether calcium detection and analysis in this paper are correct or not.

The authors defined the frequency of the oscillation as inter peak interval. However, there did not show the definition of peak. There lacks of information to define whether oscillation in each cell has equal tempo or not. Calcium oscillations usually decompose into a lot of frequency oscillations after Fourier’s transform.

Calcium dynamics is not only characterized by frequency. The authors have to analyze the amplitude of oscillations.

Taken together, authors have to rethink the analytical method, because their main focus is oscillation itself. A reviewer regarded this manuscript as a rejection.

Author Response

Reviewer 1.

Reviewer’s comment: “Torre and colleagues tried to clarify time-course Ca oscillations change in differentiation.

Their topic is original, however, there are some critical points in analysis of oscillations.

The authors did not show original oscillation wave in each cell. Readers cannot judge whether calcium detection and analysis in this paper are correct or not.”

Author’s response: We thank the reviewer for this comment. Example traces have been now added for each experimental group (see supplemental information Figure 1).

Reviewer’s comment: “The authors defined the frequency of the oscillation as inter peak interval. However, there did not show the definition of peak.”

Author’s response: The peak was defined as the maximum point of fluorescence in time. When 2 peaks were detected in sequence, the time interval between the two peaks was used to define the frequency of the event (expressed as the inverse function: the interpeak interval). In the text “The IPI was measured as the difference between the 2nd and the 1st peak by arithmetic subtraction of the respective time of the peak. (lines 182– 184)” and “The peak was the maximum point of the oscillation. Line 180”.

Reviewer’s comment: “There lacks of information to define whether oscillation in each cell has equal tempo or not.”

Author’s response: The oscillations we have detected where regular events and, in many cases, slow ones with no more than 2 peaks.

 Reviewer’s comment: “Calcium oscillations usually decompose into a lot of frequency oscillations after Fourier’s transform.”

Author’s response: We agree with the reviewer. The separation of the of the calcium wave into its components, following Fourier transform, is surely an interesting kind of analysis, but it exceeds the aims of this paper and it requires longer recordings. 

Reviewer’s comment: “Calcium dynamics is not only characterized by frequency. The authors have to analyze the amplitude of oscillations.”

Author’s response: While the amplitude may be an interesting quantity to measure, here we have used a non ratiometric probe, which does not provide accurate estimates of the amplitude. Our aim was to investigate the effect of time and treatment over the temporal dynamics of the oscillations and the fraction of cells showing oscillations.

Reviewer 2 Report

Within their manuscript authors, report changes in calcium oscillations in ASC, which were induced to differentiate either osteogenic or adipogenic, compared to undifferentiated control. Calcium imaging was performed using Fluo-4 AM and 2-photon microscopy. All in all this first data represent an interesting extension of knowledge regarding molecular mechanisms and processes of ASC characteristics and differentiation.

The authors used commercially available cells from Lonza. Did these originate from one single donor or have the experiments been repeated with cells from further donors? This should be stated as it gives the reader an idea whether oscillations might be regulated individually or whether they found evidence for an overall phenomenon.

For a general overview I would welcome that authors state the number of measurement performed for each treatment group

Did authors approve osteogenic and adipogenic differentiation also with additional markers? Studies from Winkel et al (Winkel, A., Jaimes, Y., Melzer, C., Dillschneider, P., Hartwig, H., Stiesch, M., ... & Hoffmann, A. (2020). Cell culture media notably influence properties of human mesenchymal stroma/stem-like cells from different tissues. Cytotherapy22(11), 653-668.) and others show that calcium deposition might not correlate with expression im osteogenic markers on RNA level and can depend on the culture medium used. It would be interesting whether oscillations correspond to expression of marker genes.

Author Response

Reviewer’s comment: “Within their manuscript authors, report changes in calcium oscillations in ASC, which were induced to differentiate either osteogenic or adipogenic, compared to undifferentiated control. Calcium imaging was performed using Fluo-4 AM and 2-photon microscopy. All in all this first data represent an interesting extension of knowledge regarding molecular mechanisms and processes of ASC characteristics and differentiation.

The authors used commercially available cells from Lonza. Did these originate from one single donor or have the experiments been repeated with cells from further donors? This should be stated as it gives the reader an idea whether oscillations might be regulated individually or whether they found evidence for an overall phenomenon.”

Authors’ response:  We thank the reviewer for spotting this and apologise for the omission. This information is now provided in the manuscript:

“Human ASCs from 3 nondiabetic adult donor lipoaspirates characterised at passage 1 were obtained from Lonza (Slough, UK). All ASCs have been characterised immunocytochemically and by tri-lineage differentiation assay as detailed by the manufacturer and as recommended by The International Society for Cellular Therapy [1]. All cells were used between passages 7 and 11. For biological replicates, ASCs within a range of 3 passages were used. “ (Line 109 – 114).

Reviewer’s comment: “For a general overview I would welcome that authors state the number of measurement performed for each treatment group”.

Authors’ response: The number of repeats for each measure in each respective group is reported in the results section. 

Reviewer’s comment: “Did authors approve osteogenic and adipogenic differentiation also with additional markers? Studies from Winkel et al (Winkel, A., Jaimes, Y., Melzer, C., Dillschneider, P., Hartwig, H., Stiesch, M., ... & Hoffmann, A. (2020). Cell culture media notably influence properties of human mesenchymal stroma/stem-like cells from different tissues. Cytotherapy22(11), 653-668.) and others show that calcium deposition might not correlate with expression im osteogenic markers on RNA level and can depend on the culture medium used. It would be interesting whether oscillations correspond to expression of marker genes.”

Author’s response: We did indeed test different markers using these differentiation protocols. These results have been already published this year in Bicer et al, 2020. We have now added reference to these results in our discussion. Line 319-336 in the manuscript:

“We noted that the different differentiation protocols led to the development of structural markers of differentiation. For example, we observed extracellular Ca2+ deposits positive to Alizarin Red-S for the osteogenic treatment.  2-photon imaging revealed, in a nonspecific manner, the presence of Ca2+ deposits at 7 DIV in ASCs treated with the osteogenic medium. This increase in Ca2+ deposition was paralleled by an increased expression of mRNA for osteocalcin and osteopontin, molecular markers for osteogenic differentiation [26]. However, this observation was in contrast with a previous research work, which showed no increase in mRNA for osteocalcin or osteopontin in ASCs treated with osteogenic medium (21 DIV) [28]. Such discrepancy may be due methodological differences between the experimental conditions, such as the composition of the differentiation media. This, in turn, may underlie the presence of reservoirs of bone marker mRNAs in the cytoplasm of ASCs, which get translated in presence of specific chemical and/or physical cues. Therefore, further investigation is needed to clarify the effect of osteogenic differentiation on molecular markers and its causal relationship with calcium deposition and oscillation. We have observed intracellular vacuoles positive to OilRedO, indicating adipogenic differentiation, for the sample treated with adipogenic differentiation medium, at 21DIVs. Once again, 2-photon imaging revealed presence of vacuoles already at 14 DIV in the sample treated with adipogenic medium.”

Reviewer 3 Report

Review report

This interesting study evaluated and analyzed the calcium homeostasis and its temporal changes during ASCs differentiation towards adipogenic or osteogenic lineage. unfortunately, there are several drawbacks related to the rationale of this article. I think the whole study lacks significant novelty and provides limited knowledge to this academic field.

  1. In Figure 1, the authors presented ARS stained images of ASCs after osteogenic induction. However, I found that the images did not show normal red discoloration indicating calcium deposition, moreover, the resolution is poor. I doubt that if the osteogenic induction really achieved. Could the authors provide more convincing data regarding the osteogenic differentiation of ASCs? ALP activity? Or the osteodifferentiation of ASCs should be verified using real-time PCR by checking osteogenic gene expression levels.
  2. In Figure 1c. the authors needs to be much more specific regarding the description of calcium oscillation among three different groups.
  3. Regarding the analysis of the fraction of cells having ca2+ oscillation: I think the data is confusing. There is a marked fluctuation of ca2+ oscillation even in control group without any external stimuli? Could authors provide the baseline Ca2+ dynamics in ASCs in shorter period of time, i.e., 0-30 minutes?

Overall , I think that the whole article really needs significant improvement before it can be considered for publication.

Author Response

Reviewer 3

Reviewer’s comment: “This interesting study evaluated and analyzed the calcium homeostasis and its temporal changes during ASCs differentiation towards adipogenic or osteogenic lineage. unfortunately, there are several drawbacks related to the rationale of this article. I think the whole study lacks significant novelty and provides limited knowledge to this academic field.

In Figure 1, the authors presented ARS stained images of ASCs after osteogenic induction. However, I found that the images did not show normal red discoloration indicating calcium deposition, moreover, the resolution is poor.”

Author’s response: We thank the reviewer for these remarks. However, the image shown in Figure 1 is a confocal image based on fluorescence of Alizarin Red and not a classic wide-field image (see also Schürmann M, Wolff A, Widera D, Hauser S, Heimann P, Hütten A, Kaltschmidt C, Kaltschmidt B. Interaction of adult human neural crest-derived stem cells with a nanoporous titanium surface is sufficient to induce their osteogenic differentiation. Stem Cell Res. 2014 Jul;13(1):98-110). Thus, the resulting image is not a conventional red discoloration but a representation of the Alizarin Red fluorescence image with a confocal laser scanning microscope. 

Reviewer’s comment: “I doubt that if the osteogenic induction really achieved. Could the authors provide more convincing data regarding the osteogenic differentiation of ASCs? ALP activity? Or the osteodifferentiation of ASCs should be verified using real-time PCR by checking osteogenic gene expression levels.”

Authors’ response: Again, we thank the reviewer for their remark. Indeed, all of the differentiation experiments have been conducted in parallel to experiments published recently in Bicer et al (Bicer, M., et al., Electrical Stimulation of Adipose-Derived Stem Cells in 3D Nanofibrillar Cellulose Increases Their Osteogenic Potential. Biomolecules, 2020. 10(12)) using cells from the same donors at same passages. In this study we have not only shown Alizarin Red data but also verified the differentiation using von Kossa stainings, assessment of alkaline phosphatase activity, and immunocytochemical staining against osteopontin and osteocalcin. 

This is now clearly indicated in the manuscript: Line 319-336 in the manuscript:

“We noted that the different differentiation protocols led to the development of structural markers of differentiation. For example, we observed extracellular Ca2+ deposits positive to Alizarin Red-S for the osteogenic treatment.  2-photon imaging revealed, in a nonspecific manner, the presence of Ca2+ deposits at 7 DIV in ASCs treated with the osteogenic medium. This increase in Ca2+ deposition was paralleled by an increased expression of mRNA for osteocalcin and osteopontin, molecular markers for osteogenic differentiation [26]. However, this observation was in contrast with a previous research work, which showed no increase in mRNA for osteocalcin or osteopontin in ASCs treated with osteogenic medium (21 DIV) [28]. Such discrepancy may be due methodological differences between the experimental conditions, such as the composition of the differentiation media. This, in turn, may underlie the presence of reservoirs of bone marker mRNAs in the cytoplasm of ASCs, which get translated in presence of specific chemical and/or physical cues. Therefore, further investigation is needed to clarify the effect of osteogenic differentiation on molecular markers and its causal relationship with calcium deposition and oscillation. We have observed intracellular vacuoles positive to OilRedO, indicating adipogenic differentiation, for the sample treated with adipogenic differentiation medium, at 21DIVs. Once again, 2-photon imaging revealed presence of vacuoles already at 14 DIV in the sample treated with adipogenic medium.”

Reviewer’s comment: “In Figure 1c. the authors needs to be much more specific regarding the description of calcium oscillation among three different groups.”

Author’s response: The legend has now been updated to better explain what is being represented in Figure 1C.

Reviewer’s comment: “Regarding the analysis of the fraction of cells having ca2+ oscillation: I think the data is confusing. There is a marked fluctuation of ca2+ oscillation even in control group without any external stimuli? Could authors provide the baseline Ca2+ dynamics in ASCs in shorter period of time, i.e., 0-30 minutes?”

Author’s response: This is correct: the fraction of cells showing oscillations changes with time in untreated ASCs. This is one of the central observations of the paper. That is, time in vitro has an effect on the properties of calcium oscillations in untreated ASCs. This is reported in the first part of the Results and discussed in the first part of the discussion section. As for the duration of the calcium oscillation frequency baseline, there is no need as we are not measuring the acute effect of a treatment within the same sample: all comparisons are between subjects for each factor (time and treatment). The calcium oscillation properties have been measured within a 20 minute time window, as specified in the methods section 2.4. 

Reviewer 4 Report

The manuscript “Time-dependent reduction of calcium oscillations in adipose-derived stem cells differentiating towards adipogenic and osteogenic lineage” submitted by Enrico C. Torre and coworkers reports a study in which the authors have used 2-photon live Ca2+ imaging in order to evaluate the oscillations and the frequency of such oscillations in primary cultures of adipose-derived mesenchymal stromal cells (ASCs) during osteogenic or adipogenic differentiation. Among other findings, the authors claim that adipogenic differentiation was associated with a more pronounced reduction of Ca2+ dynamics compared to cells differentiating towards the osteogenic lineage. Overall, the manuscript is interesting and well presented. However, I recommend the incorporation of additional information and experimental data.

Major comments:

  • Sections 2.2 and 2.3. The composition of the osteogenic and adipogenic differentiation media and supplements needs to be reported. This is of particular relevance in case that both media have different Ca2+.
  • The cell viability should be evaluated to identify possible differences in cells viability among all treatment groups and for all the time-points analyzed. For instance, other previous studies have found that cells viability was reduced in cells undergoing adipogenic differentiation.

Author Response

 Reviewer 4

The manuscript “Time-dependent reduction of calcium oscillations in adipose-derived stem cells differentiating towards adipogenic and osteogenic lineage” submitted by Enrico C. Torre and coworkers reports a study in which the authors have used 2-photon live Ca2+ imaging in order to evaluate the oscillations and the frequency of such oscillations in primary cultures of adipose-derived mesenchymal stromal cells (ASCs) during osteogenic or adipogenic differentiation. Among other findings, the authors claim that adipogenic differentiation was associated with a more pronounced reduction of Ca2+ dynamics compared to cells differentiating towards the osteogenic lineage. Overall, the manuscript is interesting and well presented. However, I recommend the incorporation of additional information and experimental data.

We thank the reviewer for their time and valuable comments.

Major comments:

  • Sections 2.2 and 2.3. The composition of the osteogenic and adipogenic differentiation media and supplements needs to be reported. This is of particular relevance in case that both media have different Ca2+.
  • Author’s response: We thank the reviewer for this remark. As indicated in the materials and methods section, differentiation has been conducted using commercial and established differentiation media and supplements. The commercial supplier is not providing exact information about the composition of the media (for IP reasons) but according to the information available on the internet and the literature, both basic media are α-MEM. Thus, the Ca2+.concentrations do not differ.  
  • The cell viability should be evaluated to identify possible differences in cells viability among all treatment groups and for all the time-points analyzed. For instance, other previous studies have found that cells viability was reduced in cells undergoing adipogenic differentiation.
  • Author’s response: Our concurrent, recent paper has investigated the viability of cells undergoing osteogenic differentiation [2], concluding that cell viability is not affected by osteogenic differentiation at 21 DIV.

Reviewer 5 Report

To Authors.

The paper by Torre et al., describes a time-dependent reduction of calcium oscillations in adipose-derived stem cells undergoing adipogenic and osteogenic differentiation, compared to controls.

Results might be interesting, but the functional significance of interpeak interval of Ca2+ oscillations should be hypothesized.

In the introduction, the authors state that intracellular Ca2+ is involved in “downstream signaling cascades and numerous cell functions, including proliferation, differentiation, and apoptosis”. However, they fail to hypothesize which signaling pathways might be differently involved in their three samples of cells (control ASCs, ASCs undergoing adipogenic or osteogenic differentiation). This would make the work more interesting from a functional point of view.

In my opinion, in this form, the paper is more suitable for a short communication.

Author Response

Reviewer 5

Reviewer’s comment: “The paper by Torre et al., describes a time-dependent reduction of calcium oscillations in adipose-derived stem cells undergoing adipogenic and osteogenic differentiation, compared to controls.

Results might be interesting, but the functional significance of interpeak interval of Ca2+ oscillations should be hypothesized.”

Author’s response: We thank the reviewer for their valuable comments. The interpeak interval is an equivalent measure of the oscillation frequency. The frequency of a calcium oscillation is a common parameter to identify the differentiation state and the activity of a cell-type. However, we are still missing a specific calcium code, allowing us to deconvolve the waveform quantifiable features and relate them to specific biological functions. This paper is a first step into the direction of clarifying the association between differentiation and calcium waves.

Reviewer’s comment: “In the introduction, the authors state that intracellular Ca2+ is involved in “downstream signaling cascades and numerous cell functions, including proliferation, differentiation, and apoptosis”. However, they fail to hypothesize which signaling pathways might be differently involved in their three samples of cells (control ASCs, ASCs undergoing adipogenic or osteogenic differentiation). This would make the work more interesting from a functional point of view.”

Author’s response: We have now updated the discussion to include a potential set of mechanistic correlates of the changes we have observed.

Line 407-417: “More specifically, it has also been shown that Ca2+ oscillations in human MSCs are generated by a combination of extracellular and intracellularly stored Ca2+. Extracellular Ca2+ enters the cell via the activation of a plasma membrane voltage gated Ca2+ channels, while intracellularly stored one can be released via the activation of IP3 sensitive Ca2+ channels on the endoplasmic reticulum, which can be activated as a downstream signalling event following the binding of a ligand to integrins and Gq-coupled receptors [3, 4]. Therefore, it is possible to infer that specific patterns of electrical stimulation may lead to specific differentiation fates (such as adipogenic and osteogenic) by modulating the properties of the intracellular Ca2+ dynamics. It is still to be established, however, how and if the modulation of Ca2+ waves is causative of the differentiation process or an epiphenomenon”.

Round 2

Reviewer 1 Report

Authors said the peak was defined as the maximum point. What is the definition of the maximum. Do you mean local maximum? You seemed to detect maximum as appearance. This is an example of incorrect time series analysis.

From fig.S1, oscillations did not occur regularly. Mathematical model adaptable to irregular event are necessary in oscillations analyses.

To discuss the dynamics of the oscillations, radiometry of calcium indicator(s) must be performed.

Authors added supplemental data, however, there are lack of confirmation to support their conclusion. This paper should be rejected. After adding experiments and analyses, papers are evaluated whether to be published or not.

Author Response

Reviewer’s comment: Authors said the peak was defined as the maximum point. What is the definition of the maximum. Do you mean local maximum? You seemed to detect maximum as appearance. This is an example of incorrect time series analysis.

Author’s response: We thank the reviewer for the comment. This would be the local maximum, defined as the point surrounded by lower values and with a first derivative in time approximating 0. The peaks were identified with the tool provided by Originpro. The text has been modified accordingly.

Reviewer’s comment: From fig.S1, oscillations did not occur regularly. Mathematical model adaptable to irregular event are necessary in oscillations analyses.

Author’s response: Figure S1 shows regular oscillations except for the adipogenic treatment, which is characterized by fewer cells showing oscillations. While looking into the Fourier transform analysis of these oscillations would be an interesting step, it exceeds the aims of this paper, which is a preliminary description of the relationship between differentiation, time and intracellular calcium oscillations.

Reviewer’s comment: Authors added supplemental data, however, there are lack of confirmation to support their conclusion.

Author’s response: the data we have added are not additional, but rather example traces used for quantifying the measures reported in Figure 2 and 3.

Reviewer 3 Report

The reviewer would like to congratulate the authors and expresses the significance and novelty of this manuscript.

I think the authors have already addressed all the raised concerns and points and i think the current manuscript is suitable for publication in this journal.

Author Response

Reviewer’s comment: The reviewer would like to congratulate the authors and expresses the significance and novelty of this manuscript.

I think the authors have already addressed all the raised concerns and points and i think the current manuscript is suitable for publication in this journal.

Author’s response: We thank the reviewer for their kind words and support.

Reviewer 5 Report

Torre et al. “Time-dependent reduction of calcium oscillations in adipose- derived stem cells differentiating towards adipogenic and osteogenic lineage”

In the revised version of the manuscript, the questions have been satisfactorily answered.

Some suggestions:

Line 120:  10% CO2 is correct?

Figure 1, panel B: scale bar is still missing.

Author Response

Reviewer’s comment: In the revised version of the manuscript, the questions have been satisfactorily answered.

Author’s response: We thank the reviewer for their feedback.

Reviewer’s comment: Some suggestions:

Line 120:  10% CO2 is correct?

Author’s response: Yes it is confirmed to be the correct concentration of CO2.

Reviewer’s comment: Figure 1, panel B: scale bar is still missing.

Author’s response: This has now been amended.